# Noninvasive Physical Plasma as Innovative and Tissue-Preserving Therapy for Women Positive for Cervical Intraepithelial Neoplasia

**DOI:** 10.3390/cancers14081933

**Published:** 2022-04-12

**Authors:** Julia Marzi, Matthias B. Stope, Melanie Henes, André Koch, Thomas Wenzel, Myriam Holl, Shannon L. Layland, Felix Neis, Hans Bösmüller, Felix Ruoff, Markus Templin, Bernhard Krämer, Annette Staebler, Jakob Barz, Daniel A. Carvajal Berrio, Markus Enderle, Peter M. Loskill, Sara Y. Brucker, Katja Schenke-Layland, Martin Weiss

**Affiliations:** 1Institute of Biomedical Engineering, Eberhard Karls University, 72076 Tübingen, Germany; julia.marzi@uni-tuebingen.de (J.M.); shannonlayland@yahoo.com (S.L.L.); daniel.carvajal-berrio@med.uni-tuebingen.de (D.A.C.B.); peter.loskill@uni-tuebingen.de (P.M.L.); katja.schenke-layland@nmi.de (K.S.-L.); 2NMI Natural and Medical Sciences Institute, 72770 Reutlingen, Germany; myriam.holl@student.uni-tuebingen.de (M.H.); felix.ruoff@nmi.de (F.R.); markus.templin@nmi.de (M.T.); 3Cluster of Excellence iFIT (EXC 2180) “Image-Guided and Functionally Instructed Tumor Therapies”, Eberhard Karls University, 72076 Tübingen, Germany; 4Department of Gynecology and Gynecological Oncology Bonn, University Hospital Bonn, 53127 Bonn, Germany; matthias.stope@ukbonn.de; 5Department of Women’s Health, Eberhard Karls University, 72076 Tübingen, Germany; melanie.henes@med.uni-tuebingen.de (M.H.); andre.koch@med.uni-tuebingen.de (A.K.); thomas.wenzel@student.uni-tuebingen.de (T.W.); felix.neis@med.uni-tuebingen.de (F.N.); bernhard.kraemer@med.uni-tuebingen.de (B.K.); sara.brucker@med.uni-tuebingen.de (S.Y.B.); 6Department of Pathology and Neuropathology, Eberhard Karls University, 72076 Tübingen, Germany; hans.boesmueller@med.uni-tuebingen.de (H.B.); annette.staebler@med.uni-tuebingen.de (A.S.); 7Fraunhofer Institute for Interfacial Engineering and Biotechnology, 70569 Stuttgart, Germany; jakob.barz@igb.fraunhofer.de; 8Erbe Elektromedizin GmbH, 72072 Tübingen, Germany; markus.enderle@erbe-med.com; 9Department of Medicine/Cardiology, University of California, Los Angeles (UCLA), Los Angeles, CA 90095, USA

**Keywords:** physical atmospheric pressure plasma, cervical intraepithelial neoplasia (CIN), low- and high-grade squamous intraepithelial lesions (LSILs and HSILs), Raman imaging, clinical plasma application

## Abstract

**Simple Summary:**

The treatment of cervical intraepithelial neoplasia (CIN) is still associated with the use of invasive therapeutic procedures. Although CIN 1/2 lesions show high remission rates, treatment is necessary in individual cases and noninvasive and oncologically safe therapeutic options should be available for these patients. Here, we characterized the antineoplastic properties of noninvasive physical plasma (NIPP) at the in vitro, ex vivo and in vivo levels and performed a prospective, single-armed phase-IIb trial on 20 patients with CIN1/2 (NCT03218436). NIPP-treated dysplastic cell models exhibited significant cell growth retardation due to DNA damage, cell cycle arrest and apoptosis. A tissue level analysis showed a transmucosal tissue devitalization while preserving the tissue morphology. Within 24 weeks of follow-up, treatment success was achieved in 19 (95%) participants with CIN 1/2 without peri- or postinterventional complications. Therefore, NIPP may be a sufficient treatment alternative for CIN, other mucosal dysplasia and beyond.

**Abstract:**

(1) Background: Cervical intraepithelial neoplasia (CIN) of long-term persistence or associated with individual treatment indications often requires highly invasive treatments. These are associated with risks of bleeding, infertility, and pregnancy complications. For low- and middle-income countries (LMICs), standard treatment procedures are difficult to implement and manage. We characterized the application of the highly energized gas “noninvasive physical plasma” (NIPP) for tissue devitalization and the treatment of CIN. (2) Methods: We report the establishment of a promising tissue devitalization procedure by NIPP application. The procedure was characterized at the in vitro, ex vivo and in vivo levels. We performed the first prospective, single-armed phase-IIb trial in 20 CIN1/2 patients (NCT03218436). (3) Results: NIPP-treated cervical cancer cells used as dysplastic in vitro model exhibited significant cell growth retardation due to DNA damage, cell cycle arrest and apoptosis. Ex vivo and in vivo tissue assessments showed a highly noninvasive and tissue-preserving treatment procedure which induces transmucosal tissue devitalization. Twenty participants were treated with NIPP and attended a 24-week follow-up. Treatment success was achieved in 19 (95%) participants without postinterventional complications other than mild to moderate discomfort during application. (4) Conclusions: The results from this study preliminarily suggest that NIPP could be used for an effective and tissue-preserving treatment for CIN without the disadvantages of standard treatments. However, randomized controlled trials must confirm the efficacy and noninferiority of NIPP compared to standard treatments.

## 1. Introduction

Cervical intraepithelial neoplasia (CIN) is a serious precursor to cervical cancer, which is the fourth most common cancer among women worldwide (270,000 deaths/year) and is associated with severe lifelong physical and emotional burdens due to both the disease and radical therapy [1,2]. Whereas CIN1 presents mild dysplastic lesions, CIN2 and CIN3 indicate moderate to severe dysplasia, including carcinoma in situ. Population-based studies determined the highest incidence of CIN1 with 5.1 per 1000 women in the age group between 20 and 24 years. The highest incidence rates for CIN 2 and CIN 3 were documented in the age group between 25 and 29 years with 3.8 and 4.1 per 1000 women, respectively [3]. The disease has a long precancerous phase, which enables at best the highly effective treatment of the precursors with low-invasive and easily applicable treatment options, particularly in low-income and middle-income countries (LMICs). Approximately 12 to 20% of CIN3 become invasive [4]. Therefore, current guidelines suggest the local excision of CIN3, usually by the invasive loop electric excision procedure (LEEP) or thermal approaches such as cryo- and thermal ablation [4,5]. The treatment of CIN1/2 is recommended for lesions persisting for over 24 months and special individual indications such as psychological stress, essential secondary diseases (e.g., HIV infection and immunosuppression) and the patient’s urgent wish for therapy [6]. LEEP is well established with high success rates, albeit it requires local or general anesthesia, and is associated with invasiveness, an increased risk of major bleeding, reduced fertility and pregnancy complications (e.g., premature delivery, low birthweight, and an increased rate of cesarean section) [7,8]. For LMICs, thermal approaches such as cryo- and thermal ablation are currently recommended; however, they are time consuming, the availability of the special equipment is poor, and the opportunity for individualized patient treatment is limited.

Thus, for LMICs (limited medical supplies) as well as for industrialized nations (overtreatment), there is an ongoing need for innovative, noninvasive treatment procedures, as overtreatment represents a serious problem for affected women and the health economy.

In contrast to current excisional and destructive thermal approaches, noninvasive physical plasma (NIPP) is a tissue-preserving, highly flexible and patient-specific molecular treatment modality. NIPP treatment has achieved promising superficial antineoplastic effects on several tumor entities in vitro and in vivo while preserving the integrity of deeper tissue [9,10,11]. ROS and RNS, formed by the interaction of NIPP with the ambient atmosphere, are responsible for the induction of antiproliferative and apoptotic cell mechanisms [12]. We recently established the VIO3/APC3 electrosurgical argon plasma device (Erbe Elektromedizin, Tübingen, Germany) for innovative, favorable in vivo application in patients [13,14,15]. 

In this study, we evaluated the clinical use of NIPP for the noninvasive treatment of cervical neoplasia. We characterized antiproliferative NIPP mechanisms and investigated clinically relevant tissue effects using contact- and marker-independent Raman imaging. We conducted a prospective single-armed clinical study in 20 patients with CIN1/2 lesions either persisting over 24 months or being associated with individual treatment indications such as psychological stress and the patient’s urgent wish for therapy. Based on the results of this study, prospective randomized controlled clinical trials may be conducted for CIN3 and further neoplastic diseases of mucosa in gynecology and beyond.

## 2. Materials and Methods

### 2.1. Patient Treatment

Study design: Here, we present findings from the prespecified pilot phase of a prospective, single-armed phase IIb clinical trial (NCT03218436), performed at the Department for Women’s Health, Tübingen, Germany. The work described was carried out in accordance with “The Code of Ethics of the World Medical Association” (Declaration of Helsinki) and was approved by the Ethical Committee of the Medical Faculty of the Eberhardt-Karls-University Tübingen (237-2017BO1). CIN was diagnosed by colposcopy-directed biopsy prior to investigational treatment. Patients were enrolled in the study after providing informed consent. The key inclusion criteria for study participation were premenopausal women, 18–50 years of age, histologically confirmed CIN1/2, visibility of the entire transformation zone and the entire lesion margin. The key exclusion criteria were histologically confirmed CIN3 or invasive/microinvasive disease, endocervical disease, severe inflammatory disease, other severe diseases, pregnancy. 10 out of 20 patients showed CIN1/2 lesions older than 24 months and therefore had an indication for ablative treatment. Three and seven patients showed CIN1/2 lesions between 6 and 12 months and 12 and 24 months, respectively. In these individual patients, ablative treatment was indicated due to urgent desire to have children or psychological stress and therefore patient’s compelling wish for therapy. Patients were informed about the experimental nature of NIPP treatment, for which, there was no clinical evidence for the therapeutic benefit at time of treatment. Furthermore, patients were informed about other existing and clinically well-established standard treatment options (such as laser and LEEP). A clinical examination by colposcopy and VIA (4% acetic acid and Lugol’s iodine staining) was performed to visualize CIN1/2 lesions, followed by NIPP treatment under colposcopic view (30 s/cm^2^ using VIO3/APC3 and 3.2 mm APC probes (settings: preciseAPC, effect 1), reusable silicone electrode mat). The NIPP probe was passed over the tissue in defined “brush strokes” to avoid localized heating of the tissue. The treatment was carried out on an outpatient basis and without local or general anesthesia.

Sensations of pain within 24 h following NIPP treatment were scored using a visual analog scale from 0–10. Scores of 0 and 1 were defined as “no pain”, 2–4 as “mild pain”, 5–7 as “moderate pain” and 8–10 as “severe pain”. Other side effects were recorded as free text according to the “Freiburg index of patient satisfaction” [16].

Study participants were re-assessed for histopathological and cytological remission 3 and 6 months following NIPP treatment. For this, a clinical examination by colposcopy and VIA (4% acetic acid and Lugol’s iodine staining) as well as colposcopy-directed biopsy was performed by trained and certified gynecologists. According to guidelines an endocervical curettage was not obtained due to complete visibility of the transformation zone and the lesion margin.

### 2.2. Physico-Chemical Characterization

Infrared thermography, electron spin resonance (ESR) spectroscopy and Optical emission spectroscopy (OES) were performed as previously described [13,17].

### 2.3. Propagation and In Vitro NIPP Treatment of Cells

Cervical squamous cell carcinoma-derived cancer cells (CaSki (ATCC CRL-1550), DoTc2-4510 (ATCC CRL-7920), SiHa (ATCC HTB-35) were purchased from ATCC (ATCC^®^ TCP-1022™, American Type Culture Collection (Manassas, VA, USA); and propagated as described previously [18].

For NIPP treatment, the VIO^®^ 3, APC 3 electrosurgical device (Erbe Elektromedizin, Tübingen, Germany) was used (argon gas flow: 1.6 L/min; preciseAPC, effect 1). 5.0 × 10^3^ cancer cells in 700 μL of DMEM were NIPP dynamically treated for 30 s in suspension in a 6-well cell culture plate in 700 µL of DMEM at a distance of 7 mm and were incubated in 6-well cell culture plates (2 mL DMEM/well) for indicated timepoints. The controls were treated with argon gas alone (flow: 1.6 L/min) to exclude any alterations in cells and tissues due to the treatment procedure.

### 2.4. Cell Proliferation

Cell number (using a CASY Cell Counter and Analyzer Model TT, Roche Applied Science, Basel, Swizzerland), cell viability (using Guava ViaCount Reagent for Flow-Cytometry, Merck Milipore, Burlington, MA, USA) and apoptosis (using a luminescent Caspase-Glo 3/7 assay, Promega, Walldorf, Germany) were analyzed as previously described [17].

### 2.5. Immunfluorescence Microscopy

Confocal laser-scanning microscopy (LSM) was carried out using an LSM 710 inverted confocal microscope (Carl Zeiss, Oberkochen, Germany). For imaging, a 405 nm laser diode and a 488 nm argon laser were used for excitation as merges of a Z-stack, spanning the extent of the cellular fluorescence signal. The cells were incubated in 6-well glass bottom cell culture plates (Ibidi, Gräfelfing, Germany, 2 mL DMEM/well). The cells were fixed with 0.25% paraformaldehyde for 15 min and stained according to specific protocols. 

Ki67: anti-Ki67-specific antibody, ab15880; Abcam, 1:1000, overnight at 4 °C; goat anti-mouse immunoglobulin G (IgG) 1, 488, Invitrogen (Carlsbad, CA, USA) secondary antibody, 1:250, 60 min. γH2AX: fixed cells were incubated in 0.1% NaN_3_ in PBS for 48 h followed by staining with anti-γH2AX-specific antibody, JBW301, Merck Millipore, 1:500, 2 h; goat anti-mouse immunoglobulin G (IgG) 1488, Invitrogen secondary antibody, 1:250 for 60 min.

5mC staining: To analyze the global genomic 5mC methylation status, we performed IF staining using a 5mC-specific mouse monoclonal IgG antibody (MABE146, 1:2000, Merck Milipore) according to a previously described protocol [19].

DAPI (1:2, in DPBS, 10 min, Roche Diagnostics, Basel, Swizzerland) was used for counterstaining. Image analysis was performed with ZEN blue software (Carl Zeiss). For every independent experiment (*n* = 3), cells from 3 different areas of the cell culture plate were evaluated.

### 2.6. Flow Cytometry

Flow cytometry was performed as previously described [20]. Applied specific antibodies:DSB-specific γH2AX formation: anti-γH2AX antibody, Ser139, JBW301, Sigma-Aldrich (St. Louis, MO, USA), 1:125 dilution, 30 min on ice. Cell cycle phase analysis: DAPI, Sigma-Aldrich, 1:2 dilution, 30 min on ice. Forward- and side-scatter (FSC-H and SSC-H) characteristics were used to exclude dead cells. Forward scatter area and height (FSC-A and FSC-H) characteristics were used to exclude cell doublets (Appendix A).

### 2.7. Quantitative Reverse Transcription-Polymerase Chain Reaction (RT-PCR)

RT-PCR was performed as previously described [11]. mRNA amounts were quantified using sequence-specific oligonucleotides: GADD45a forward: 5′-TCGTGAAATGGAAGGGATGG-3′, GADD45a reverse: 5′-AGGTTTTGGGCTTGGGTC-3′; Reference gen ribosomal protein L19 (RPL19) forward: 5′-ATGAGTATGCTCCGGCTGCAG-3′, RPL19 reverse: 5′TCACTTCTTGGTCTCTTCTTC-3′.

PCR was carried out with Sensimix SYBR Hi-Rox (Bioline, London, UK) in a QuantStudio 3 Detection System (Applied Biosystems, Waltham, MA, USA). For quantification, GADD45 signals were standardized to RPL19 mRNA as a reference, and the fold change was calculated according to the 2^ΔΔCT^ method.

### 2.8. DigiWest Multiplex Protein Profiling 

After a 30 s NIPP treatment of 5.0 × 10^3^ cervical cancer cells in 700 μL of DMEM, the cells were incubated in 6-well cell culture plates (2 mL DMEM/well) for the indicated time periods. Analysis was performed by DigiWest multiplex protein profiling, as described previously [21]. The following primary antibodies were used: pH 3-specific antibody: 9701, Cell Signaling Technology(Danvers, MA, USA), 1:200; Cyclin B1-specific antibody: ab32053 (Y160), Abcam, 1:200; protein kinase B (pAKT)-specific antibody: 12,178 (D5G4), Cell Signaling Technology, 1:200; heat shock protein 27 (HSP27)-specific antibody: 2402 (G31), Cell Signaling Technology, 1:200; Bcl2-family member Bim (Bim)-specific antibody: 2819, Cell Signaling Technology, 1:200; Caspase 9 (Casp9)-specific antibody: 9502, Cell Signaling Technology, 1:200; p53 phosphorylation (pp53)-specific antibody: 9284, Cell Signaling Technology, 1:200; p53 binding protein 1 (53BP1)-specific antibody: 4937, Cell Signaling Technology, 1:200.

### 2.9. Raman Microspectroscopy and Raman Imaging

Raman measurements of single cells and tissue samples were performed as previously described [15]. The scientific use of the tissue was approved by the Ethical Committee of the Medical Faculty of the Eberhardt-Karls-University Tübingen (649-2017BO2).

### 2.10. Histology

Lactate dehydrogenase (LDH) assay: NIPP treated fresh tissue samples were cryopreserved with TissueTek (Sakura Finetek, Staufen im Breisgau, Germany) and frozen at −80 °C before being sectioned into 10 µm cross-sections. The coverslips were processed according to Hess R. et al. [22]. *HE staining.* Routine histological hematoxylin and eosin staining was performed according to standard protocols.

### 2.11. Statistical Analysis

Statistical comparisons were carried out with Student’s t test or ANOVA (GraphPad Prism version 6.0, GraphPad Software, San Diego, CA, USA), as specified in the figure legends. The data are expressed as the mean ± standard deviation. *p* values of <0.05 were considered to indicate statistical significance.

## 3. Results

### 3.1. In Vitro NIPP Treatment and Molecular Analysis of the Human Malignant Cervical Cancer Cells

NIPP is capable of inducing complex biological effects, including antineoplastic efficacy due to ROS and RNS, which are formed at the interface of NIPP and the environmental gas and liquid phases of the treated tissue [23]. Before safely and noninvasively applying NIPP, we deeply characterized the tissue surface temperature by infrared thermography, the formation reactive species by ESR and the emission of energy by fully integrated OES using an integrating sphere (Appendix A).

Next, the antineoplastic efficacy of NIPP could be demonstrated in vitro by utilizing human malignant cervical cancer cells. NIPP-treated cancer cells exhibited a significant decrease in cell number and cellular activity (Figure 1a,b). This was accompanied by significantly altered gene methylation patterns 4 h after NIPP treatment (Figure 1c,d). Twenty-four hours after treatment, this alteration was statistically significant but clearly less pronounced.

Due to the chemical properties of ROS and RNS, which frequently interact with nucleic acids, NIPP treatment is followed by the induction of complex cellular DNA damage recognition and repair mechanisms [24]. A sensitive biomarker for DNA damage, particularly DNA double-strand breaks, is the phosphorylation of histone H2AX at Ser139 [25]. We showed that NIPP-induced redox stress resulted in rapid but transient H2AX phosphorylation. The signals of phosphorylated histones significantly increased within 1 h and 4 h after NIPP treatment and decreased again within the following 20 h of incubation (Figure 1e–g). DNA double-strand breaks were followed by the activation of DNA repair mechanisms, including elevated expression levels of growth arrest- and DNA damage-associated gene 45 (GADD45) and poly(ADP-ribose)-polymerase (PARP) (Figure 1h).

Cell cycle analysis revealed a transient and slight increase in the G2-phase population 4 h after NIPP treatment (*p* = 0.08). This may indicate that an increased number of cells arrested in the G2 phase and did not enter mitosis. However, the effect was identical after 24 h (*p* = 0.99, Figure 1i). Further protein analysis demonstrated a significant increase in cyclin B1 expression, indicating G2 arrest (Figure 1j). This was accompanied by a crucial decrease in the mitotic biomarker phospho-histone H3 (Ser10) [26,27].

Apoptosis, which is most likely the physiological consequence of severe DNA damage and cell cycle arrest, was induced in cells via the activation of the effector cysteine-dependent aspartate-specific proteases (caspases) casp3 and casp7 (Figure 1k). Interestingly, at the investigated time point, the initiator caspase casp9 was already found to be downregulated again (Figure 1l). Furthermore, the expression of pivotal antiapoptotic factors, such as heat shock protein 27 (HSP27) and protein kinase B (AKT), was suppressed. In contrast, a set of proapoptotic factors was induced by NIPP treatment: Bcl-2 homology 3 domain-only protein (BIM), phosphorylated p53 (pp53) and p53-binding protein 1 (53BP1).

Single-cell Raman spectroscopy combined with principal component analysis (PCA) (Appendix A) confirmed DNA-related NIPP effects (Figure 1m), whereas alterations in lipid composition occurred instantly (Figure 1n).

### 3.2. Ex Vivo NIPP Treatment and Molecular Analysis of Cervical Tissue Samples

As the neoplastic transformation of nonmalignant keratinocytes into premalignant cells takes place in the basal cell layer, any topical treatment for CIN must pass the superficial layer to affect the neoplastic cells within the basal layer.

To determine whether NIPP treatment can pass the entire cervical epithelium, we obtained cervical tissue samples from six patients undergoing hysterectomies for other indications (Appendix A). Immediately after surgical removal, the tissue samples were treated with NIPP ex vivo and analyzed by histochemical methods as well as contact-free Raman imaging (Appendix A). Areas of devitalized tissue could be detected by a loss of lactate dehydrogenase (LDH) activity following NIPP treatment (Figure 2a, upper row). LDH-negative tissue areas increased in a dose-dependent manner with increasing NIPP treatment duration. However, we found no impact on tissue morphology after hematoxylin-eosin (HE) staining (Figure 2a, lower row), highlighting the noninvasive nature of NIPP.

Raman imaging enabled the identification and localization of tissue-specific structural components (nuclei, collagens, cytoplasmic proteins) in cervical tissue (Appendix A). The application of Raman imaging to NIPP-treated cervical tissue confirmed the morphological integrity and revealed no significant differences (Figure 2b,c). In-depth molecular-level analysis of nuclear features was performed by multivariate analysis. PCA of nuclear signals demonstrated statistically significant dosage- and time-dependent differences compared to nontreated tissue (Figure 2d). The observed effects were predominantly detected in superficially located cells; however, they were also detectable in cells of the basal tissue layer.

### 3.3. In Vivo NIPP Treatment of Patients with CIN

Next, for in vivo evaluation, NIPP was applied in 20 patients with histologically proven CIN in a prospective, single-armed phase IIb clinical trial (NCT03218436) at the Dysplasia Center of the Department for Women’s Health, Tübingen, Germany (Figure 3a). CIN was diagnosed by colposcopy-directed biopsy prior to NIPP treatment. The key inclusion criteria for study participation were premenopausal women, 18–50 years of age, histologically confirmed CIN1 or 2, visibility of the entire transformation zone and the entire lesion margin. Key exclusion criteria included CIN3 or invasive disease, endocervical involvement, signs of severe inflammation, pregnancy and severe secondary diseases. Seven patients (35%) were diagnosed for CIN1, 13 patients (65%) for CIN 2. Indication for treatment was given either by persisting lesions over 24 months (10 patients, 50%) or by the patient’s compelling wish for therapy (urgent desire to have children, great fear and/or psychological stress). As precursors of the target lesion CIN3, CIN1/2 was used as preliminary in vivo model to evaluate the potential to treat CIN3. Before NIPP treatment the patients gave their written informed consent to the ethically approved study protocol (237-2017BO1) and were informed about the experimental nature and other existing and clinically well-established standard treatments (such as laser and LEEP). The CIN lesions were visualized by colposcopy and VIA, followed by NIPP treatment under colposcopic view (without local or general anesthesia) (Figure 3b–d and Appendix A). Histopathological and cytological remission was assessed after 12 and 24 weeks.

Among the patients enrolled, an additional colposcopy-directed tissue biopsy was taken from 6 patients immediately after NIPP treatment. The cryopreserved tissue samples were analyzed by Raman imaging (for patient characteristics of this subgroup refer to Appendix A).

Raman imaging revealed highly comparable molecular signatures, as shown in Figure 2b–d. The human cervix consists of the squamous (ectocervix) and columnar (endocervix) epithelium, and our data showed that NIPP treatment similarly affected both types of tissue (Figure 3e–j and Appendix A). The treatment effects could be identified in superficial and basal epithelial layers, indicating the transmucosal efficacy of NIPP (Appendix A). Taken together, these data demonstrate that NIPP affects all epithelial layers. Despite the identified molecular changes, tissue morphology was preserved after in vivo NIPP treatment (Figure 3e,h).

Across all 20 patients who received NIPP treatment, no acute dose-limiting toxicities were observed. A total of 66.7% of the patients (*n* = 12) experienced mild adverse events (grade 1), including smear bleeding (27.8%, *n* = 5) and increased vaginal discharge (38.9%, *n* = 7), within 24 h posttreatment. The most common of these events was local discomfort, including pain and cramping, during NIPP treatment, all of which were mild (50.0%, *n* = 9) to moderate (22.2%, *n* = 4) in severity. All adverse events resolved spontaneously without treatment.

Colposcopy-directed biopsy 12 and 24 weeks after NIPP treatment demonstrated that 95% (*n* = 19) of CIN1/2 patients achieved complete histological remission (Figure 4a,b). PAP smear tests performed 2, 12 and 24 weeks after NIPP treatment showed cytological normalization of formerly suspicious results. One CIN2 patient (5%) showed partial histological remission to CIN1 after 12 weeks, followed by complete remission after 24 weeks. One former CIN1 patient (5%) with initial complete remission after 12 weeks showed progression to an CIN2 12 weeks later. Due to the explicit wish of the patient, LEEP was performed 28 weeks after NIPP treatment. The final histological result confirmed a 2 mm CIN1.

## 4. Discussion

In 2018, the World Health Organization (WHO) initiated a global call for action to eliminate cervical cancer, including strategies for HPV vaccination, screening and effective treatment of women with precancer or cancer [28]. Ninety-one percent of cervical cancer deaths occur in LMICs due to insufficient coverage with HPV vaccination as well as screening and treatment of CIN. In the present study, we report the first application of NIPP in humans to treat CIN1/2 using a next-generation electrosurgical argon plasma device. The use of NIPP is a novel and innovative concept for anesthesia-independent, tissue-preserving and easy performable cancer prevention.

Conventional therapy for HSILs is commonly based on local excision of the lesion via LEEP including local or even general anesthesia. LEEP is still associated with considerable tissue damage. This may lead to severe bleeding, reduced fertility rates and complications during pregnancy and delivery (e.g., premature delivery, low birthweight, and an increased rate of cesarean section) [7,8]. The number of possible repetitions is limited due to the considerable loss of tissue mass. 

Additional strategies to treat CIN include thermal approaches involving the application of a low temperature (cryoablation, −70 °C or lower) or high temperature (heat ablation, at least 100 °C) to induce necrotic tissue destruction. Cryotherapy and nowadays also thermal ablation are WHO recommended ablative treatments for LMICs. A major limitation of cryotherapy is the need for a refrigerant gas (N_2_O or CO_2_), which is for the most bulky, heavy to transport, costly and difficult to supply. Advantages of thermal ablation are the simple and lightweight equipment and the possibility to perform treatment without anesthesia.

In contrast to the application of NIPP (at 30 s per cm^2^), both thermal approaches are rather time-consuming [29]. Cryoablation requires up to 11 min per treated area (3 min of freezing, 5 min of thawing, and 3 min of freezing). Occasionally, thermal ablation requires multiple overlapping applications [29]. Both thermal approaches are restricted to the dimensions of commercially available and predefined thermoprobes, whereas NIPP treatment is highly flexible and individually adaptable to the size and shape of the lesion. Furthermore, technically immature devices as well as increasing costs have limited the implementation of thermal approaches in clinical routines [30]. A disadvantage of all thermal ablation procedures—such as as NIPP treatment—is that no definitive histology can be obtained, which necessitates control examinations. A currently ongoing trial by Pinder et al. compares the efficacy of heat ablation, cryoablation and LEEP (NCT02956239) [29]. The treatment success rate in the pilot phase, defined as either HPV type-specific clearance and/or negative visual inspection, was reported as 60% for cryoablation, 64% for heat ablation and 67% for LEEP. In contrast to strictly physical procedures of tissue destruction, NIPP induces specific molecular cell responses. These responses primarily target cell growth, metabolism, the cell cycle and apoptosis [31]. Our previous analyses have shown that nonthermally operated NIPP devices lead to the formation of ROS [13], which are primarily responsible for the observed clinical effects, including antiproliferative and antineoplastic effects, on human cervical cancer cells [14,15].

In the present study, the tissue-preserving nature of NIPP was demonstrated. NIPP induced a devitalization of all mucosal tissue layers, even under low-dose treatment, which was detectable at the molecular and cellular levels. At the single-cell level, we observed the effects of NIPP on gene expression and cell proliferation (Figure 5). In this context, transient DNA damage was associated with downstream cell cycle arrest and apoptosis.

Despite substantial formation of ROS, suggesting chemical modifications of DNA molecules, no genotoxic effects of NIPP have been demonstrated thus far [32]. In the case of DNA damage, either immediate DNA repair or apoptosis is initiated [33]. Cell cycle analysis and particularly the downregulation of mitosis-specific histone H3 phosphorylation, have shown that a subset of the cell population remains in the G2 phase [27]. Supported by the induction of GADD45, acting as an inhibitor of cyclinB1 (which subsequently triggers G2 arrest) [34], this suggests, that cell cycle arrest occurs after NIPP-induced DNA damage. If DNA repair is no longer possible, apoptotic pathways are induced, which is a well-known mechanism of action during NIPP treatment [35].

One single in vivo application of NIPP was followed by cytological and histological complete remission of CIN in 19 out of 20 patients (95%) within 6 months. One patient (5%) showed progression to CIN2 within the follow-up period. Due to its highly flexible and patient-specific application, NIPP enabled side effect-free and nearly painless treatment of CIN1/2 without any postintervention complications. Moreover, the procedure was accompanied by maximum cervical tissue preservation. The NIPP procedure was carried out using a modular high-frequency device, which is highly available in surgical facilities. A down-sized single-mode and battery-driven generator powered by solar panels could extend the use of this new therapeutic approach to LMICs.

Currently, different treatment modalities are recommended for cervical neoplasia depending on the severity and time of persistence. According to most national guidelines CIN 1/2 can be managed in a watch and wait approach due to the rate of spontaneous remission (40–60%). Treatment is only recommended for lesions persisting over 24 months in addition to special individual situations [36]. CIN3 remains the actual target lesion for immediate treatment. Here, we used CIN1/2 as preliminary in vivo model to evaluate the potential for CIN3 treatment. Depending on upcoming prospective, randomized and controlled studies, NIPP may also be indicated for the treatment of CIN3 in the future with significantly less invasiveness and better tolerability. Furthermore, NIPP could be a suitable treatment for individual cases of CIN1/2which may reduce the cancer risk-related psychological burden of the typically young patients. Beyond cervical neoplasia, NIPP technology could be of interest in cases of other mucosal neoplasia, e.g., those located in the gastrointestinal tract, lungs, vagina, and larynx.

## 5. Conclusions

NIPP treatment results in tissue-preserving cross-mucosal tissue penetration and devitalization. The efficacy of NIPP is most likely explained by the rapid but transient DNA damage, with consequent inhibitory impacts on the cell cycle, metabolism and cell proliferation. This leads to the initiation of the apoptotic machinery. The ability to noninvasively cure CIN1/2 in vivo indicates the potential of NIPP application for the well-tolerated treatment of CIN3 and other neoplasia.

## Figures and Tables

**Figure 1 cancers-14-01933-f001:**
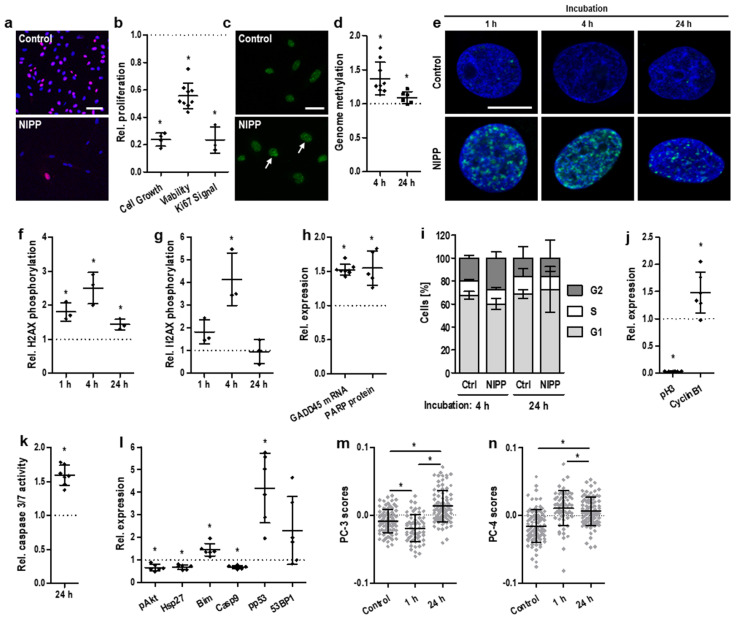
In Vitro NIPP treatment induces antiproliferative cell effects at different molecular interfaces. (**a**) Representative IF staining of Ki67 72 h after 30 s of NIPP treatment and (**b**) relative antiproliferative efficacy 72 h after 30 s of NIPP treatment), shown as relative decreases in cell number, cell viability, and Ki67 expression; the scale bar equals 40 μm. (**c**) Representative IF staining of 5mC 4 h after 30 s of NIPP treatment and (**d**) relative genomic methylation level per nucleus (number of foci normalized to the control) 4 (◆) and 24 h (■) after 30 s of NIPP treatment of cervical cancer cells; the scale bar equals 20 μm. (**e**) Representative IF microscopy of γH2AX and (**f**) relative γH2AX intensity and (**g**) relative γH2AX intensity 1 h, 4 h, and 24 h after 30 s of NIPP treatment of cervical cancer cells, indicating DNA double-strand breaks; the scale bar equals 10 μm. (**h**) Relative induction of growth arrest- and DNA damage-inducible genes GADD45 and PARP 72 h after 30 s of NIPP treatment. (**i**) Relative fractions of cells in 561 cell cycle phases S, G1 and G2 and with reversible phase transition to G2 4 h after 30 s of NIPP treatment. (**j**) Relative expression of the cell cycle factors phospho-histone H3 (Ser10) and CyclinB1 72 h after 30 s of NIPP treatment, indicating G2 arrest. (**k**) Relative caspase-3/7 activity 24 h after 30 s of NIPP treatment. (**l**) Relative expression of the apoptotic factors pAkt, Hsp27, Bim Casp9, pp53, and 53BP1 24 h after 30 s of NIPP treatment. (**m**) PC-3 and (**n**) PC-4 score values for single-cell Raman microspectroscopy of untreated and 30 s NIPP-treated cervical cancer cells after 1 and 24 h, as assessed by PCA; the data for each group originate from 3 independent experiments with 30 cells each. Results are expressed as mean ± SD; (**a**–**l**): * *p* < 0.05; paired *t* test; (**m**,**n**): one-way ANOVA; * *p* < 0.05).

**Figure 2 cancers-14-01933-f002:**
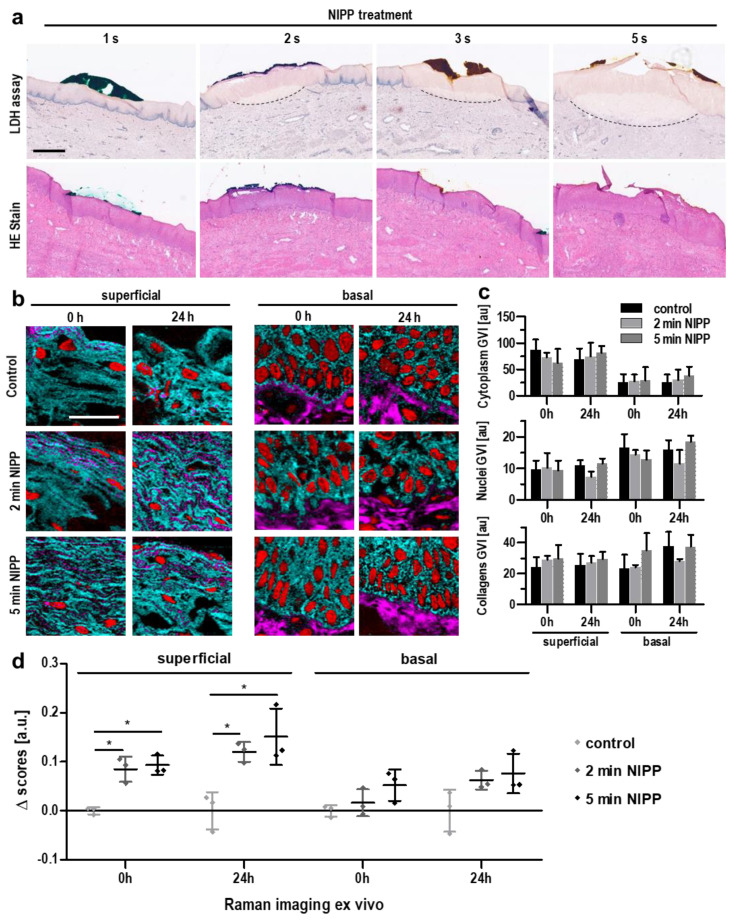
Ex vivo NIPP treatment maintains tissue morphology but shows transmucosal penetration and immediate effects on cell physiology. (**a**) Representative lactate dehydrogenase (LDH; upper row) and hematoxylin-eosin (HE, lower row) staining of cervical tissues after treatment with NIPP at different doses; the scale bar equals 200 μm. (**b**) Raman intensity distribution heatmaps assigned to collagen I (pink), nuclei (red) and cytoplasmic proteins (light blue) immediately and 24 h after ex vivo NIPP treatment of cervical tissue for 2 and 5 min; the scale bar equals 50 μm. Images were acquired 583 from the basal and superficial tissue layers. (**c**) The gray value intensities (GVIs) of the Raman images assessed in (**b**) revealed no quantitative differences between treated and untreated tissues. (**d**) Statistical comparison of the nuclear spectra obtained in (**b**) was performed by PCA and subsequent normalization of the PC score values to the control samples to assess qualitative differences in nuclear composition; the data points represent average score values per donor (*n* = 3). Results are expressed as mean ± SD; two-way ANOVA; * *p* < 0.05).

**Figure 3 cancers-14-01933-f003:**
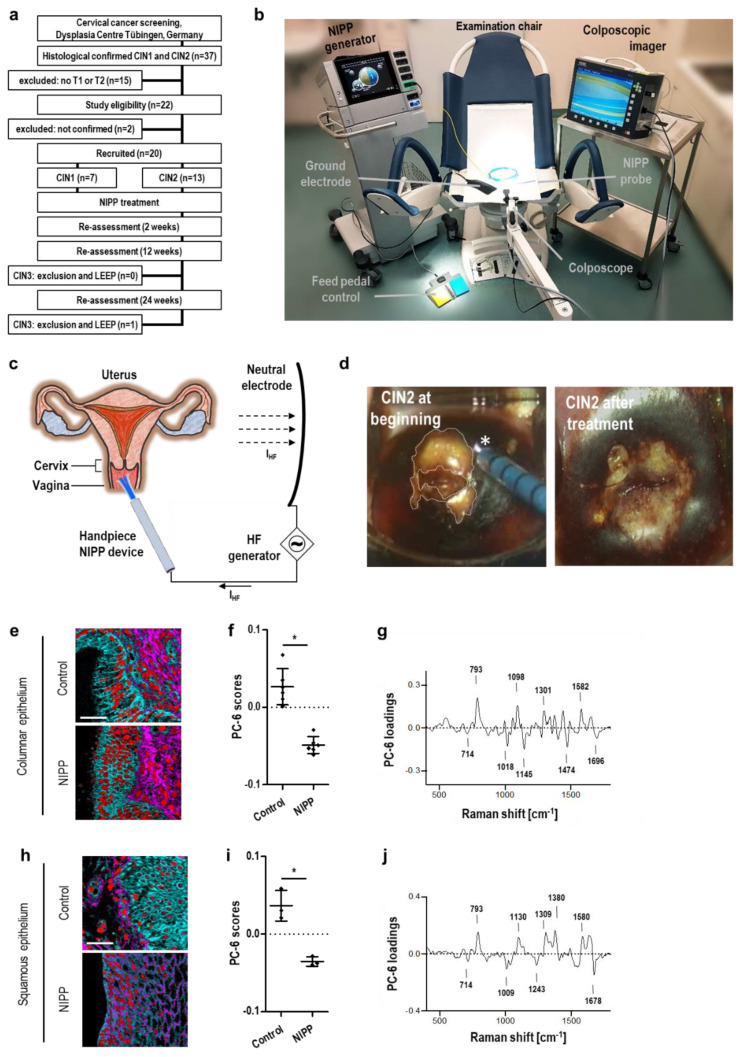
In vivo NIPP treatment and evaluation of molecular tissue effects, penetration depth and clinical efficacy. (**a**) Trial flow chart of patient recruitment and study visits for the prospective clinical study. (**b**) Setup of in vivo NIPP treatment. (**c**) Schematic of the anatomical view of the human cervix. As the uterine portion of the vaginal cavity, the cervix is easily accessible for NIPP treatment. (**d**) Colposcopic image of a human cervix after staining with 4% acetic acid and Lugol’s iodine at beginning (**left**) and after (**right**) NIPP treatment (30 s/cm^2^). The transformation zone is completely visible (T1 transformation zone; dashed line). The NIPP effluent is marked by an asterisk. (**e**–**j**) Columnar (**e**–**g**) and squamous (**h**–**j**) epithelium of tissue sections from patients treated with NIPP in vivo at 30 s/cm^2^ and tissue biopsies before treatment (control) were analyzed by Raman imaging. (**e**,**h**) Nuclei (red), collagen I (pink) and cytoplasmic proteins (light blue) were localized by TCA; the scale bar equals 50 μm. (**f**,**i**) The nuclear spectra obtained in (**e**,**h**) were processed by PCA, and the average PC score values of each patient (◆) were statistically compared (mean ± SD; paired *t* test; * *p* < 0.05). (**g**,**j**) Underlying biochemical information was interpreted based on relevant spectral signatures elaborated in the loading plot.

**Figure 4 cancers-14-01933-f004:**
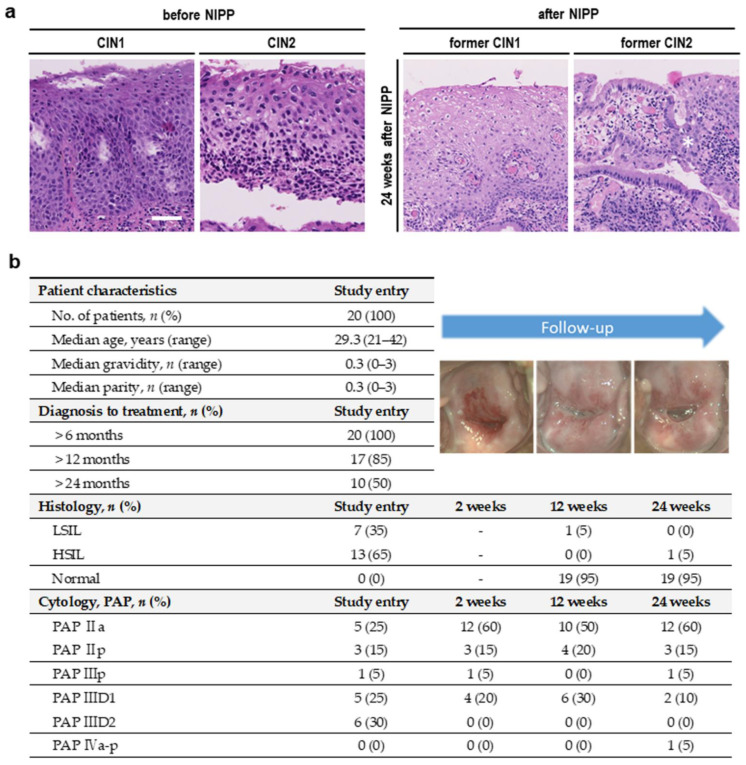
Study results. (**a**) Representative histopathological images of a CIN1 a CIN2 lesion before and 24 weeks after NIPP treatment within the prospective clinical study; the scale bar equals 50 μm; the asterisk marks the junction between squamous and columnar epithelium. (**b**) Clinical, histopathological and cytological features of patients before and 2, 12 and 24 weeks after NIPP treatment.

**Figure 5 cancers-14-01933-f005:**
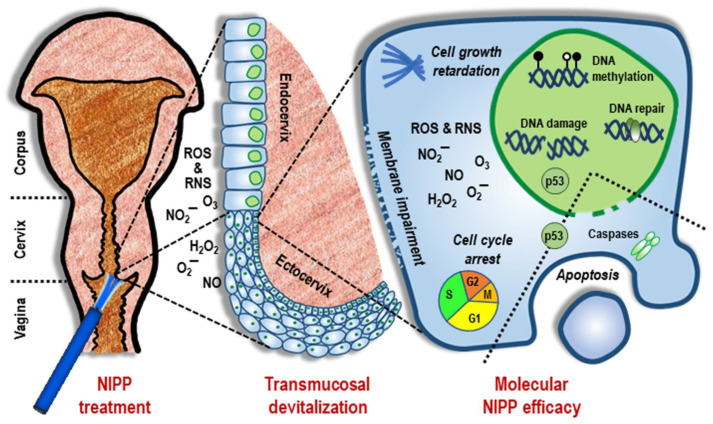
General summary of the antineoplastic NIPP response. The reactive species generated by NIPP cause a transmucosal increase in intracellular ROS and RNS, particularly through NIPP-mediated impairment of the cytoplasmic membrane. As a result, ROS and RNS activate various intracellular response pathways, primarily alterations in genomic methylation patterns and signal transduction cascades involved in the DNA double-strand break response and p53-associated apoptosis. This is followed by the attenuation of cell growth, arrest of the cell cycle, and the initiation of apoptosis.

## Data Availability

The data presented in this study are available on request from the corresponding author. The data are not publicly available due to the ongoing character of the entire research project.

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
