# Peer review of "Noninvasive Physical Plasma as Innovative and Tissue-Preserving Therapy for Women Positive for Cervical Intraepithelial Neoplasia"

_cancers, 2022, doi:10.3390/cancers14081933_

Round 1

Reviewer 1 Report

The authors present an interesting manuscript using non-thermal physical plasma for the non-invasive treatment of cervical intraepithelial neoplasia. The methodology is thorough and well described. The plasma treatments are well characterized, as are the effects through in-vitro, ex-vivo and in-vivo treatments. The conclusions are thus well founded in the experimental results, and are highly relevant for cancer prevention.

One minor point, in the published manuscript, it would be recommendable to stick to standardized phrasing that can be most easily picked up through machine recognition. - For the Author Contributions, this is presented by the CRediT (Contributor Roles Taxonomy)  system (c.f. https://onlinelibrary.wiley.com/doi/epdf/10.1002/leap.1210), which is hence recommended to be utilized.

Author Response

We want to thank the reviewers for their constructive comments and suggestions, which clearly helped to further improve the work. Please find our detailed point-by-point responses below.

Response to the reviewer:

Reviewer 1, Comment 1: One minor point, in the published manuscript, it would be recommendable to stick to standardized phrasing that can be most easily picked up through machine recognition. - For the Author Contributions, this is presented by the CRediT (Contributor Roles Taxonomy) system (c.f. https://onlinelibrary.wiley.com/doi/epdf/10.1002/leap.1210), which is hence recommended to be utilized.

- We thank reviewer 1 and agree that it is essential to use standardized phrasing. However, we strictly used the system recommended by Cancers to describe Author Contributions. Due to this we did not change the section Authors contributions at first and would like to wait for the editor’s opinion on this.

Reviewer 2 Report

Title - It is better to remove an abbreviation from the title.

Abstract - represents the study very well. However, the first sentence in the abstract is disputable. It should be corrected. NOT all CIN cases require invasive treatment. The authors should be more specific. Please refer to ASCCP guidelines.

The introduction represents the study rationale very well.

In the material and methods, please provide information on the study design and the study subjects before the other details (not in 2.10 but 2.1).

More details about the statistical analysis performed are required.

The results are interesting and presented with great details and supported by high-quality images and figures. 

According to the classical academic writing style, a sentence should not begin with numbers, therefore, in line 380 "91%" should be spelled as "Ninety-one percent".

Is it necessary to supply the conclusion with the figure? It would be better to keep it in the discussion part and make the conclusion more concise.

Author Response

We want to thank the reviewers for their constructive comments and suggestions, which clearly helped to further improve the work. Please find our detailed point-by-point responses below. 

Response to the reviewer:

Reviewer 2, Comment 1: Title - It is better to remove an abbreviation from the title.

- We thank reviewer II and agree with his comments. Thus, we have changed the title to:

Non-invasive Physical Plasma (NIPP) as Innovative and Tis-sue-preserving Therapy for Women Positive for Cervical in-traepithelial Neoplasia

Reviewer 2, Comment 2: Abstract - represents the study very well. However, the first sentence in the abstract is disputable. It should be corrected. NOT all CIN cases require invasive treatment. The authors should be more specific. Please refer to ASCCP guidelines.

- We agree with Reviewer II and thank for this comment. In several other passages in the manuscript – especially in the “Materials and Methods” section – we define the indications of CIN I/II treatment in detail. Furthermore, and following the comment of Reviewer II we adapted the first sentence of the abstract as follows:

Cervical intraepithelial neoplasia (CIN) of long-term persistence or associated with individual treatment indications often require highly invasive treatments.

- We referred to the ASCCP guidelines by citing the following publication:

>> R. B. Perkins, R. S. Guido, P. E. Castle, D. Chelmow, M. H. Einstein, F. Garcia, et al. 2019 ASCCP Risk-Based Management Consensus Guidelines for Abnormal Cervical Cancer Screening Tests and Cancer Precursors. J Low Genit Tract Dis 2020 24 102-131<<

Due to the fact, that citations shouldn’t be included into the Abstract, we inserted the citation within the following sentence:

“Treatment of CIN1/2 is recommended for lesions persisting over 24 months and special individual indications such as psychological stress, essential secondary diseases (e.g. HIV infection and immunosuppression) and the patient's urgent wish for therapy [6].”

Reviewer 2, Comment 3: In the material and methods, please provide information on the study design and the study subjects before the other details (not in 2.10 but 2.1).

- We thank Reviewer II for his comment and have adjusted the Materials and Methods section accordingly.

Reviewer 2, Comment 4: More details about the statistical analysis performed are required.

- We thank Reviewer II for his comment. Due to the variety of different methods used in this manuscript, we feel that a collected listing in section 2.11. would either lead to reader confusion or unnecessary duplications within the manuscript. Therefore, we have included method-specific statistical details in the relevant sections describing each method. We also refer to this procedure in section 2.11.

The details about statistical analysis are therefore described and/or referred to if already published.

If possible, we would therefore like to refrain from changing the current section 2.11.

Reviewer 2, Comment 5: According to the classical academic writing style, a sentence should not begin with numbers, therefore, in line 380 "91%" should be spelled as "Ninety-one percent".

- We adjusted the concerning passage as recommended.

Reviewer 2, Comment 6: Is it necessary to supply the conclusion with the figure? It would be better to keep it in the discussion part and make the conclusion more concise.

- As Reviewer II has suggested, we have included Figure 5 in the discussion section.